**www.cambridge.org/ext**

# Global threat status, rarity, and species distribution affect prevalence of Atlantic Forest endemic birds in citizen-collected datasets

biodiversity hotspots; Brazil; citizen science; community science; monitoring

**Corresponding author:**
Judit K. Szabo;
Email: judit.szabo@cdu.edu.au

Lucas Rodriguez Forti[1,2], Ana Marta P. R. da Silva Passetti[2], Talita Oliveira[3], Juan Lima[4], Arthur Queiros[3], Maria Alice Dantas Ferreira Lopes[5] and Judit K. Szabo[6]

[1]Departamento de Biociências, Universidade Federal Rural do Semi-Árido, Av. Francisco Mota, 572 – Bairro Costa e Silva, 59625-900, Mossoró Rio Grande do Norte, Brazil; [2]Programa de Pós-Graduação em Ecologia: Teoria, Aplicações e Valores, Instituto de Biologia, Universidade Federal da Bahia, Rua Barão de Jeremoabo, 668 – Campus de Ondina CEP: 40170-115 Salvador Bahia, Brazil; [3]Undergraduate program in Ecology, Universidade Federal Rural do Semi-Árido, Av. Francisco Mota, 572 – Bairro Costa e Silva, 59625-900, Mossoró Rio Grande do Norte, Brazil; [4]Programa de Pós-Graduação em Ecologia e Conservação, Universidade Federal Rural do Semi-Árido, Av. Francisco Mota, 572 – Bairro Costa e Silva, 59625-900, Mossoró Rio Grande do Norte, Brazil; [5]Undergraduate program in Veterinary Medicine, Universidade Federal Rural do Semi-Árido, Av. Francisco Mota, 572 – Bairro Costa e Silva, 59625-900, Mossoró Rio Grande do Norte, Brazil and [6]Research Institute for the Environment and Livelihoods, Charles Darwin University, Casuarina, Northern Territory 0909, Australia

## Abstract

The Atlantic Forest is one of the most threatened biomes globally. Data from monitoring programs are necessary to evaluate the conservation status of species, prioritise conservation actions and to evaluate the effectiveness of these actions. Birds are particularly well represented in citizen-collected datasets that are used worldwide in ecological and conservation studies. Here, we analyse presence-only data from three online citizen science datasets of Atlantic Forest endemic bird species to evaluate whether the representation of these species was correlated with their global threat status, range and estimated abundance. We conclude that even though species are over- and under-represented with regard to their presumed abundance, data collected by citizen scientists can be used to infer species distribution and, to a lesser degree, species abundance. This pattern holds true for species across global threat status.

## Impact statement

Given the high rates of biodiversity loss globally, knowledge gaps need to be filled urgently in order to inform and prioritise conservation actions. Research and conservation are particularly important in tropical and megadiverse biomes, such as the Atlantic Forest. Given the lack of resources available for scientific research, professional scientists are struggling to conduct studies in these fragile biomes, particularly on long-term and at large scales. However, in the past two decades, nonprofessional scientists have been participating in the research process. Furthermore, data collected by these actors have been used in large-scale studies. Therefore, citizen science is becoming an important player in biodiversity knowledge production. Nevertheless, spatial, temporal and other biases resulting from unstructured sampling need to be understood and accounted for in order to make the collected data useful for decision-making. In this study, we evaluate how the estimated abundance, global threat status and spatial distribution of species affect the number of observations citizen scientists collect. We use endemic Brazilian Atlantic Forest bird species occurrence data from three online citizen science platforms. We found that threatened species were less frequently observed by citizen scientists than non-threatened species. Species with larger distribution ranges had more observations than species with more restricted ranges in all global threat status categories. In conclusion, citizen science data can be used to predict species distribution ranges, reducing knowledge gaps for Brazilian Atlantic Forest birds. Therefore, considering data contributed by citizen science can shorten the path to conservation actions.

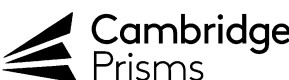



## Introduction

Studies of the biodiversity of the Brazilian Atlantic Forest biome have resulted in important datasets of morphological traits and species abundance (Hasui et al., 2018; Rodrigues et al., 2019). Nevertheless, there is still a large Wallacean deficit with regard to the biodiversity of the biome (Colli-Silva et al., 2020; Marques and Grelle, 2021). Species endemism is exceptionally high in the

Atlantic Forest (Costa et al., 2000; da Silva et al., 2004; Cruz and Feio, 2007) and given the long history of deforestation (Dean, 2002) and the effects of climate change (Vale et al., 2018), this biome has been classified as one of the most threatened biodiversity hotspots on the planet and its exuberant flora and fauna are a constant source of concern for conservation biologists. In spite of ongoing restoration efforts (Romanelli et al., 2022), many endemic species still face a high risk of extinction (de Lima et al., 2020). Large data gaps plague up-to-date estimations of population size, dynamics and distribution of most species, making threat status assessments and conservation action prioritisation inaccurate. Threatened species tend to be rare and have a more restricted distribution than species evaluated as Least Concern on the global Red List by the International Union for Conservation of Nature (https://www.iucnredlist.org/). Species facing a higher risk of extinction often require broader actions and more intensive monitoring than less threatened taxa (Green and Young, 1993; Martikainen and Kouki, 2003).

Birds in particular are highly threatened in the Atlantic Forest – five to seven bird species have likely been driven to extinction in the wild in this biome and a further nine are Critically Endangered (Develey and Phalan, 2021). Fortunately, this group is also popular among observers, as besides paid scientists, 30–40,000 Brazilian birdwatchers are known to generate information for bird conservation (Develey, 2021).

Unlike "traditional" science, which is conducted by highly trained and paid personnel, community (or citizen) science data are contributed by volunteer members of the public (Louv and Fitzpatrick, 2012). These initiatives have become instrumental in generating monitoring data globally and at multiple scales (Chandler et al., 2017). In addition, observations often originate from private properties and other areas, which are not always accessible to professional researchers (Callaghan et al., 2021). Crowdsourcing through digital citizen science platforms has increased the rate of global biodiversity information production (Kelling et al., 2019). The number of occurrence records collected by volunteers within global databases, such as the Global Biodiversity Information Facility (GBIF; https://www.gbif.org/), has massively increased over time (Boakes et al., 2010; Petersen et al., 2021). Birds, in particular, have received the most interest from citizen scientists all around the world, and consequently, this group has the highest representation within GBIF (Troudet et al., 2017).

Currently, over 60% of all GBIF species occurrence data from Brazil are birds recorded through eBird (ebird.org). eBird accepts lists, photos or sound recordings of birds that observers see or hear while walking transects or through incidental observations. Observers also record geographic coordinates and the time and the day of the observation. The eBird database is curated by taxon specialists and the platform provides scientists and the interested public with real-time data on bird distributions and abundance. Other platforms have also become popular in Brazil and produce data on the distribution of birds in the country. For instance, WikiAves is a Brazilian website for birdwatchers, with the objective of supporting, disseminating and promoting birdwatching activities through photos and sound recordings, while helping with the identification of species and encouraging communication between observers. WikiAves accepts photos and sound recordings of bird species that occur in Brazil, but does not require exact coordinates of the observations, only the name of the municipality where the bird was recorded. Among other topics, this database has been used to study species distribution and migration (Cunha et al., 2022; Atwood, 2023), behaviour (Tubelis and Sazima, 2020; de Souza

et al., 2022; Tubelis et al., 2022), diet (Schneider et al., 2023), and species interactions (Bosenbecker et al., 2023). A third platform with a high number of bird observations (270,888) in Brazil is iNaturalist. This generalist platform accepts photos and sound recordings of any organism and has been an important source of biodiversity data (Seregin et al., 2020; Mesaglio and Callaghan, 2021; Forti and Szabo, 2023). In this platform, artificial intelligence suggests identification for the submitted images and other users, including taxon experts, also contribute with their knowledge. Providing exact geographic coordinates makes it possible to use iNaturalist observations in a wide range of scientific studies, enabling spatial analyses and inferring relationships between organisms and their habitats, climate and other characteristics (Forti et al., 2022a, 2022b).

The increasing number of occurrence records collected by citizen scientists reflects a combination of increased public awareness and participation in citizen science initiatives and new technologies for recording and submitting observations (Chandler et al., 2017; Mihoub et al. 2017). At the same time, the mobilisation of other data sources, such as museum collections and the published literature has also increased the number of occurrence records in GBIF (Boakes et al. 2010). The combination of these data sources has allowed robust studies in the area of biogeography and macroecology (Liu et al., 2022; Moles and Xirocostas, 2022; Martinez et al., 2023).

In ecological studies, the number of observed individuals is often used as a proxy for species abundance. However, observations submitted by volunteers are often biased spatially – more frequent in areas of high human population density (Di Cecco et al., 2021; Forti and Szabo, 2023), temporally – observers prefer months and days when they are free and when climatic conditions are favourable (Bowler et al., 2022) and by taxonomy – depending on species characteristics, such as body colour, size, and shape (Callaghan et al., 2021; Marcenò et al., 2021). Also, the behaviour and habitat preference of some species make recording them more difficult, demanding higher observer skills or more experience, and this can result in the underrepresentation of some species of conservation concern in citizen science datasets and numbers of observations that do not directly reflect true abundance (Szabo et al., 2012). In spite of these issues, unstructured data from eBird, WikiAves and iNaturalist have been used to support conservation decision-making (Schubert et al., 2019; Spear et al., 2023). In this context, understanding factors that affect the number of observations submitted by citizen scientists is important during data analysis and interpretation.

In this work, we study the relationship between the distribution (extent of occurrence) and estimated abundance and biomass of species with the number of observations made by citizen scientists. We focus on endemic birds of the Brazilian Atlantic Forest using data from three major citizen science platforms. We also evaluate the relationship between species distribution and the number of observations in relation to the global threat category. While these relationships may seem intuitive, the behaviour of observers can vary between regions and the composition of different observer profiles can change the patterns of the data collected by them (Tulloch and Szabo, 2012). Nevertheless, our hypothesis is that the number of bird observations in the datasets is a function of the threat status of the species, which, in turn, is affected by species rarity and trends, reflected by the extent of geographic distribution and the abundance of the species (IUCN, 2022). We also describe under-, and overrepresented species and list potential actions to fill

knowledge gaps, particularly species occurrence and population trends, of Brazilian Atlantic Forest bird species. In addition, we suggest future directions for the use of citizen science data in biodiversity conservation in this highly threatened biome.

## Results

We identified a total of 1,204,210 observations of endemic birds from the Atlantic Forest that have been submitted by citizen scientists to the three platforms. After removing duplicate observations and restricting the dataset to 2000–2022, 838,880 observations remained, representing approximately 70% of the raw data.

We found positive correlations between the range of species distribution, their extent of occurrence and their estimated abundance (see raw data in Supplementary Table 1). The size of the distribution range and threat status of the species affected the number of observations submitted by citizen scientists (Figure 1). The first mixed model (AIC = 3835.317; $r^2$ = 0.50; REML = 255.6) had a positive value for the estimated coefficient (estimate = 0.21073; t-value = 5.312; p < 0.01) for the number of observations due to the range of species distributions, even considering the effects of family and threat status as random factors. These two random factors retained a large proportion of the variation in the residuals, and the value for threat status ($SD_{threat}$ = 0.3027) was higher than the value for family ($SD_{family}$ = 0.1473) with $SD_{residual}$ = 0.4054. Nevertheless, the interaction between range and threat status was not significant in their effect on the number of observations (Supplementary Table 2). A larger distribution range seemed to result in more observations within threat categories (Figure 2). Based on the interaction term, IUCN status did not significantly influence the slope of the range size,

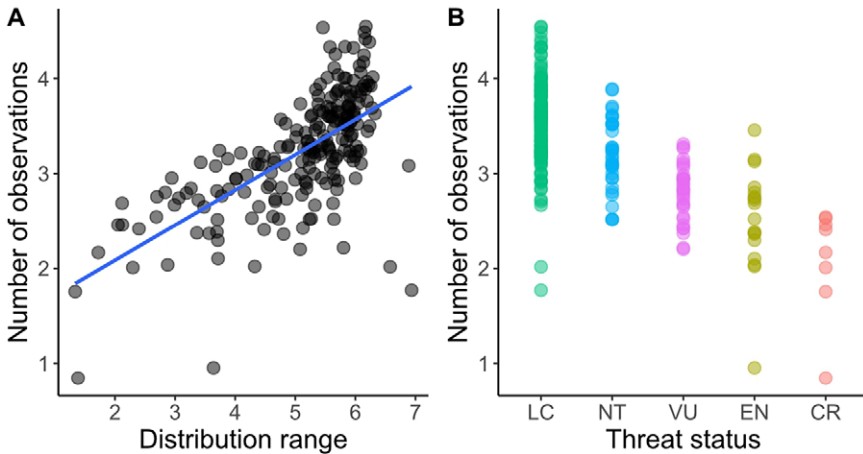

**Figure 1.** Number of observations of bird species endemic to the Atlantic Forest in Brazil in three citizen science platforms according to the distribution range of species (A) (both variables in log10 scale); and (B) the global threat status of the species (IUCN 2023).

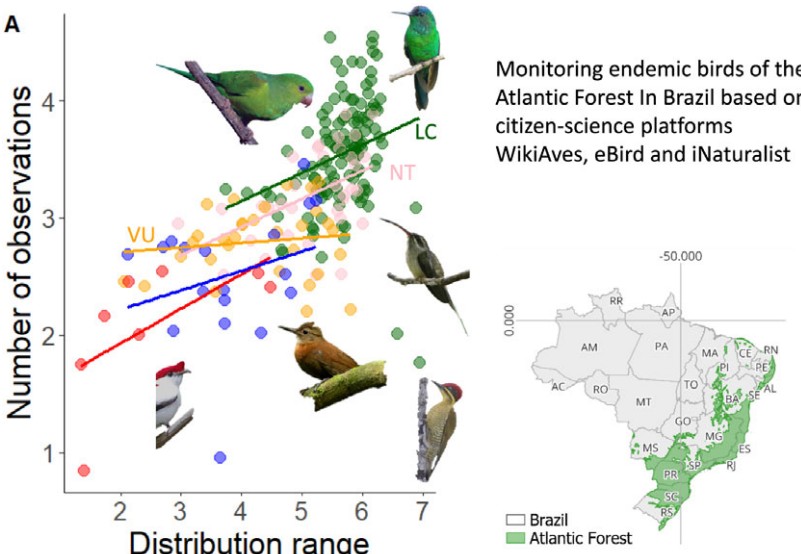

**Figure 2.** Number of observations of birds in the Atlantic Forest carried out by citizen scientists in relation to the distribution range of the species (both variables in log10 scale). Regression lines were calculated based on the global threat categories (IUCN): LC: Least Concern, NT: Near Threatened, VU: Vulnerable, EN: Endangered and CR: Critically Endangered. Species illustrated at the bottom of the graph are under-represented, such as the critically endangered *Merulaxis stresemanni* and *Antilophia bokermanni*, the Vulnerable *Sclerurus cearensis* and the Least Concern *Phaethornis malaris*. The species illustrated at the top of the graph, *Brotgeris tirica* and *Thalurania glaucopis* are overrepresented in the database. Images were provided by the following iNaturalist observers: @Anderson Sandro, @Luiz Alberto Santos, @Nereston Camargo, @Tomaz Melo, @Douglas Clarkee and @manequinho.

hence the positive relationship held true across threat categories. In the second mixed model, estimated total biomass also had a significant effect on the number of observations made by citizen scientists (REML = 87.9, estimated coefficient = 0.209, and p = 0.004). However, based on a visual analysis of the residuals and the results of the

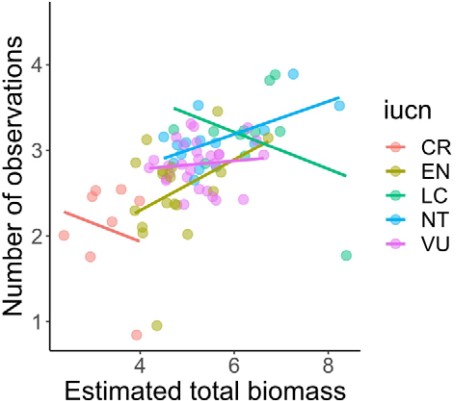

**Figure 3.** Number of observations of birds in the Atlantic Forest carried out by citizen scientists in relation to the estimated total biomass (both variables in log10 scale). Regression lines were calculated based on the global threat categories (IUCN): LC: Least Concern, NT: Near Threatened, VU: Vulnerable, EN: Endangered, and CR: Critically Endangered.

third mixed effect model for estimated total biomass controlled only by IUCN status, it had a worse fit than the previous model (REML = 106.8, estimated coefficient = 0.1379, $r^2$ = 0.329, and p = 0.018), with different patterns for different threat categories. The effect was negative for Critically Endangered and Least Concern species and positive for the rest, i.e., higher estimated biomass led to more observations (Figure 3).

Some Least Concern species, such as the Golden-green Woodpecker (*Piculus chrysochloros*; r = − 1,878) and the Great-billed Hermit (*Phaethornis malaris*; r = − 1,674; Supplementary Table 1) deviated from the model prediction by having negative residuals and were under-represented in the citizen science data. Certain threatened species also had lower than predicted representation, including the Endangered Boa Nova Tapaculo (*Scytalopus gonzagai*; r = − 1,284); and the Critically Endangered Araripe Manakin (*Antilophia bokermanni*; r = − 0.998). On the other hand, common Least Concern species, such as the Plain Parakeet (*Brotogeris tirica*) and the Ruby-crowned Tanager (*Tachyphonus coronatus*) were overrepresented, both with r = 0.892. Some threatened species were also overrepresented in the dataset, such as the Critically Endangered Orange-bellied Antwren (*Terenura sicki*; r = 0.352), the Endangered Vinaceous-breasted Amazon (*Amazona vinacea*; r = 0.464) and the Vulnerable Fork-tailed Tody-tyrant (*Hemitriccus furcatus*; r = 0.475). Feeding habits and behaviour of the species did not explain model deviations and did not directly affect the number of observations per species in the datasets (Figure 4).

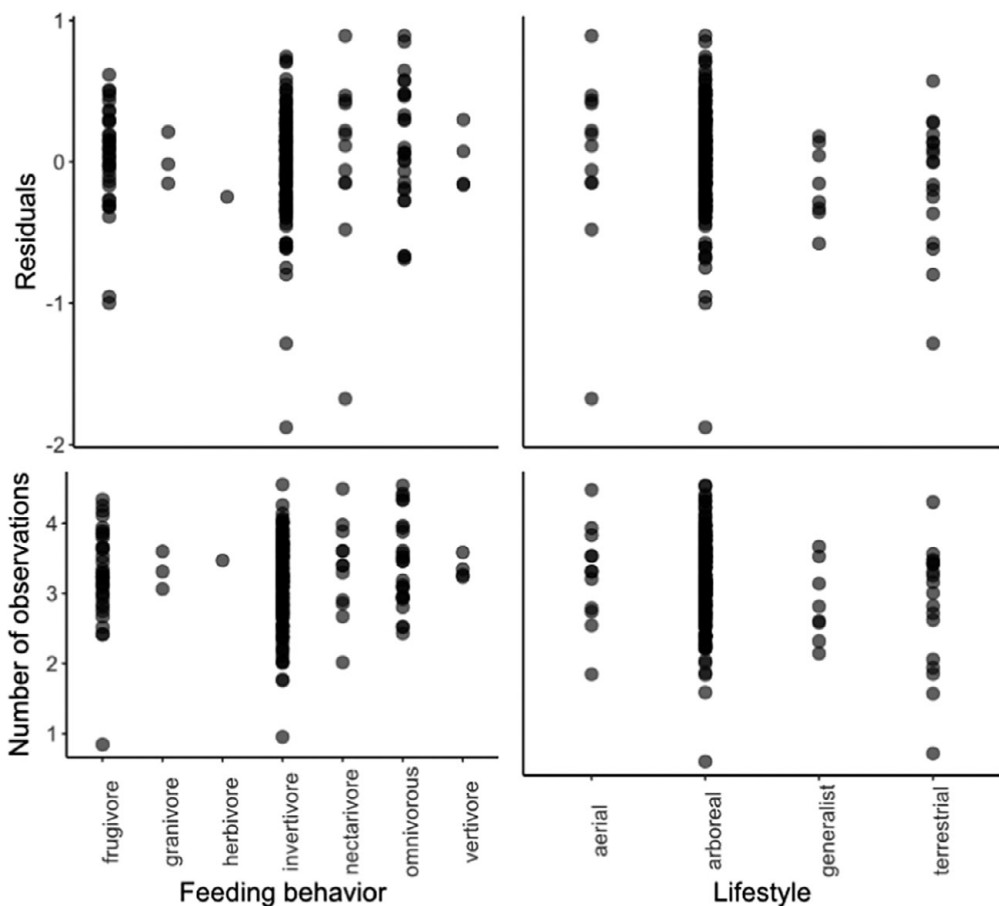

**Figure 4.** Number of observations made by citizen scientists of birds with different feeding behaviour (top left) and vertical strata (top right) and the distribution of model residuals for different categories of feeding behaviour (bottom left) and vertical strata (bottom right) for Atlantic Forest endemic birds. The number of observations and the residuals are shown at a logarithmic scale.

## Discussion

Our results suggest that threatened species are less frequently observed by citizen scientists in the Brazilian Atlantic Forest than nonthreatened species. Specifically, the more threatened a species is, the fewer observations the database contains. Based on our model, this pattern is due to range size and potentially, to a lesser extent, population size. This pattern suggests that citizen science can provide useful data for assessing the population status of birds in the Atlantic Forest, mainly with regard to the distribution range of the species. Therefore, citizen science data can reflect changes in the spatial distribution of bird species in the Atlantic Forest.

In fact, 67% of threatened and Near Threatened species of the total 216 Atlantic Forest bird species reported in the three citizen science datasets show declining population trends, while only 5% have unknown trends (BirdLife International, 2023). Given that citizen scientists can have a preference for rare or threatened species as shown in data from Australia (Tulloch and Szabo, 2012), it is not always possible to use the number of observations as an approximation of the size or distribution of bird populations. Furthermore, dull-coloured and shy species can be underreported by casual observers (Szabo et al., 2012).

Another caution we must take when interpreting citizen science data is related to oversampling in urban areas – either due to increasing spatial sampling bias over time or environmental change pooled with constant spatial sampling bias. This bias can lead to an overestimation of declines in species that are negatively affected by urban cover (Bowler et al., 2022). This effect may also explain the worse performance of the alternative model that incorporated total biomass based on species abundance estimates. Many factors could affect the number of observations, and these effects can be stronger than the abundance of the species.

Studies assessing observer behaviour have also shown a taxonomic bias in the representativeness of species in citizen science datasets (Tulloch and Szabo, 2012; Callaghan et al., 2021). In fact, body size has been an important predictor of detectability, with larger animals seen more often than smaller ones (Callaghan et al., 2021). However, some threatened species had a relatively high representation in our dataset regardless of their body sizes and these species are known to be charismatic and the object of organised initiatives (Martinez and Prestes, 2021). One example is the Vinaceous-breasted Amazon (Zulian et al., 2021), which has been reintroduced into the wild in some areas and has been a coveted target for observers. Local and focal citizen science projects have also been successful, particularly those involving iconic species, such as the Toco Toucan (*Ramphastos toco*; Schaaf et al., 2024).

In general, habitat loss in the Atlantic Forest makes continued sampling by citizen scientists even more important. Within 20 to 30 years, unstructured databases are estimated to gain more importance as their use in population trend calculations will increase (Szabo et al., 2010). Gathering information from different data sources can help to separate species dynamics from spatial biases in sampling (Dorazio, 2014; Fithian et al., 2015; Pacifici et al. 2017). This information can support the simultaneous modelling of presence-only data and standardised or presence-absence data in integrated distribution models (Dorazio, 2014; Fithian et al., 2015; Pacifici et al. 2017). With protection, habitat recovery and restoration, ongoing monitoring is more important than ever (Develey and Phalan, 2021). Up-to-date population sizes and ranges can inform us whether these actions are sufficient, or whether other measures, such as predator control, translocations, or *ex situ*

management need to be applied (Develey and Phalan, 2021). Bird observation has improved the attitude of the Brazilian public towards biodiversity, promoting bird conservation and increasing knowledge about the birds of Brazil (Develey, 2021).

Barriers to participating in citizen science have decreased over the past decade due to new outreach projects and smartphone apps, leading to greater inclusion of people with less experience, and this has apparently been happening at the national scale in Brazil (Forti and Szabo, 2023). Recently joined participants may differ in their recording behaviour and be less likely to visit remote places to record species compared to observers who have been active for decades, but even so, data contributed by people of different profiles are important to detect trends and monitor changes in species distribution. High public engagement in citizen science is crucial and initiatives involved in adaptive sampling that addresses spatial and temporal gaps, as well as taxonomic bias need to be supported (Callaghan et al., 2023). Educational projects using a citizen science approach can be particularly important in collecting data in under-sampled regions (Forti, 2023).

As databases grow, we continue to learn about biases and errors in citizen science data, including identification errors (Gorleri and Areta, 2022; Gorleri et al., 2023). Nevertheless, the resulting large databases have created enormous opportunities for ecologists to address questions about biodiversity patterns at large spatial scales (Theobald et al. 2015). Developments in statistical modelling also enable us to explain many of the biases and sources of heterogeneity in unstructured data (Isaac et al. 2014). The lack of standardised long-term monitoring for most taxa also increases the value of these datasets when assessing species turnover in ecological communities over time.

When planning citizen science initiatives, sample representativeness should be maximised (Callaghan et al., 2023). Local residents (as opposed to visitors) should be encouraged to survey the birds, as repeated surveys add value to monitoring also this type of volunteer is known to visit "less exciting" locations and record common species (Tulloch and Szabo, 2012). Citizen science data can also be better integrated with structured surveys conducted by professional scientists, in the sense that monitoring through standardised surveys should focus on the gaps left by volunteers (Tulloch et al., 2013).

## Conclusions

As the observation patterns identified for Atlantic Forest endemic birds might not be representative of all taxonomic groups in this biome, further studies should focus on the contribution of citizen science to observations of other taxa at a large scale. Our results suggest that citizen science initiatives contribute to our knowledge about endangered species in the biome in a meaningful way and this approach is expected to become even more relevant in the future for decision-making involving rare and/or threatened species.

## Methods

Study Area: Our study was conducted in the Atlantic Forest, which is the second largest tropical forest in South America behind the Amazon. The Atlantic Forest extends along the entire Brazilian coast and contains large human population centres, such as São Paulo, Rio de Janeiro and Salvador (Marques and Grelle, 2021). A complex topography covers a wide range of elevations from sea level to almost 3000 m a.s.l. and different substrates contribute to

an intricate vertical stratification, creating microhabitats for a highly diverse biota (Morellato et al., 2000; Ramalho, 2004). The vegetation of this biome is a complex of evergreen, deciduous and semi-deciduous forests, along with mangroves, dunes and high-altitude fields (Ribeiro et al., 2011). These characteristics resulted in centres of endemism for multiple taxa and made the Atlantic Forest highly biodiverse (da Silva and Casteleti, 2003; Cardoso da Silva et al., 2004). Given the large extension of the Atlantic Forest, many areas are still poorly studied (Marques and Grelle, 2021). Starting with the Portuguese colonisation of Brazil, almost 500 years ago, anthropogenic pressures reduced the extent of native vegetation in the Atlantic Forest to 7.6% of its original extent (Marques and Grelle, 2021). Deforestation rates were historically driven by clearing for sugar cane and coffee plantations (Dean, 2002). Although habitat destruction has slowed down, climate change and the fragmentation of forest remnants still represent a major threat to the biodiversity of this biome (SOS Mata Atlântica/INPE, 2018; de Lima et al., 2020). In spite of many recent reforestation initiatives (https://pactomataatlantica.org.br/), endemic bird species are still declining (Develey and Phalan, 2021). The extent of protected areas is also relatively low, only covering 2% of the original area with native vegetation (Tabarelli et al., 2005).

Proceedings: To evaluate the representativeness of citizen science data, we extracted metadata from the three most important citizen science platforms in Brazil for 216 endemic birds of the Atlantic Forest (Vale et al., 2018). For species taxonomy, we followed BirdLife International (2023). The list of species and their status are detailed in Supplementary Table 1. We obtained data through formal requests to the Application Programming Interface (API) of eBird (https://ebird.org/home) and iNaturalist (https://www.inaturalist.org/), and compiled metadata using the Instant Data Scraper application (https://webrobots.io/instantdata/) from WikiAves (https://www.wikiaves.com.br/index.php). From iNaturalist, we only considered "research grade" observations, i.e., records validated at the species level with a consensus from at least 2/3 of the identifiers. We manually obtained population trends from BirdLife International's Data Zone (2023). In the case of birds, BirdLife International is also the official assessor of IUCN Red List status. We obtained the IUCN global threat classification of species through the *rredlist* package (Chamberlain, 2020) in R version 4.2.1 (R Core Development Team, 2022). We included species from the following categories: Extinct (EX) for species, where there is no reasonable doubt that the last individual has died; extinct in the wild (EW), for species considered extinct in their natural habitat; Critically Endangered (CR), Endangered (EN), and Vulnerable (VU), following quantitative criteria designed to reflect varying degrees of threat of extinction (taxa in any of these three categories are collectively referred to as "threatened" henceforth); Near Threatened (NT), which is applied to species that currently do not meet the criteria for threatened, but are close to it or are likely to become threatened if ongoing conservation actions are reduced, interrupted or ceased; and Least Concern, for species that do not qualify (and are not close to qualifying) as threatened or Near Threatened. The category Least Concern indicates that, in terms of extinction risk, these species are of lower concern than species in other threat categories and does not imply that these species are not of conservation concern. None of the birds of the Atlantic Forest were classified by IUCN as Data Deficient or Not Evaluated. Five quantitative criteria are used to determine the threat category of a particular species, based on biological indicators of threatened populations, such as rapid population decline or reduced population size. These five criteria are as follows:

A. Population size reduction (past, present and/or projected);
B. Size of geographic distribution and fragmentation, few locations conditioned to threat, decline or fluctuations;
C. Small population size with decline and fragmentation, fluctuations or few subpopulations;
D. Population size too small or distribution too narrow;
E. Quantitative Extinction Risk Analysis (e.g., Population Feasibility Analysis)

We obtained minimum and maximum abundance estimates and the extent of occurrence (EOO) of each species from BirdLife International (https://www.birdlife.org/datazone). As another metric, we obtained the size of the distribution range, which indicates the total area of the mapped range for the species from AVONET (Tobias et al., 2022). These numbers were calculated based on BirdLife International maps, considering areas, where a particular species was coded as present and distinguishing native and reintroduced ranges and areas, where the species was resident or visitor. We also collated data on body mass, feeding behaviour and life history traits (arboreal, aerial, etc.) for each of the species based on Tobias et al. (2022).

The area of occurrence (AOO) represents the geographical range of a species, which is calculated using a minimum convex polygon based on observation locations (Gaston, 1991). This metric is essential to evaluate a taxon based on Criterion B and can be used in Criterion A, which is used to assess changes in the distribution of a species (IUCN, 2022).

Data analysis: Statistical analyses and graphical visualisations were produced using R version 4.2.1 (R Core Development Team, 2022). We checked the heterogeneity of the dataset (abundance pattern based on the number of observations by each species) by applying Benford's Law (Szabo et al., 2023) and found it to have marginal conformity with regard to the distribution of digits, which means that the data are of satisfactory quality (Forti et al., 2024). We produced graphs using the *ggplot2* package (Wickham, 2016).

We excluded one species, the Alagoas Curassow (*Mitu mitu*), from all analyses. Until its recent reintroduction in 2019 (Francisco et al., 2021), this species had the previous (unconfirmed) sighting in the wild in the late 1980s and is still considered EW by BirdLife International (2023). While our citizen-collected dataset contained two observations, there are no population size or range estimates provided by BirdLife International (2023).

As IUCN status is calculated over three generations or 20 years, we limited our database to observations made between January 1, 2000 and December 31, 2022. We also eliminated duplicate observations of the same species that occurred at the same geographic location on the same day using the function distinct from the *dplyr* package (Wickham et al., 2022). To obtain a more realistic estimate of species abundance, we calculated the median from the minimum and maximum values obtained from the Data Zone interface of BirdLife International (2023). We calculated total biomass by multiplying the median estimate of species abundance by the body mass of the species (Tobias et al., 2022).

We used three generalised mixed models through the *lme4* (Bates et al., 2015) and *lmerTest* (Kuznetsova et al., 2017) packages to assess the effect of species abundance, biomass and EOO, and distribution on the number of observations submitted by citizen scientists. First, we checked the correlation between fixed factors before fitting the models to avoid collinearity and log10 transformed all numerical variables to eliminate discrepancies in the dataset. Since the data were not normally distributed, we used the *cor.test* function through Spearman's method to correlate

the logarithm of the median abundance with the logarithm of the EOO and the distribution of the species. As the logarithm of the distribution range was highly correlated with species abundance and the EOO (Supplementary Table 3), we included it as a fixed factor in the first model, assigning the logarithm of the number of observations as the dependent variable and family and IUCN status as random factors. We used these two variables as random factors because both were correlated with the distribution range and estimated total biomass (Supplementary Table 4). In the second mixed model, we fit base 10 logarithm of the total estimated biomass of the species as a fixed factor and the same condition as in the first model for the random factors and the dependent variable. Then, we fit a third mixed model to isolate the effects of biomass on the number of observations controlled by IUCN status, using IUCN status as a random factor. No singular fit problems were identified for these models. We assessed the normality of the residuals visually using the *qqnorm* and *qqline* functions. After fitting the models, we obtained the residual maximum likelihood value (REML) and annotated the estimates of each fixed effect, as well as their significance value. We calculated variation around the estimates using a 95% confidence interval through the *confit.merMod* function. We checked indices of model performance and singularity using the *model_performance* and *check_singularity* functions of the *performance* package (Lüdecke et al., 2021). We saved residuals from the first mixed model to identify species that were under and overrepresented in the database. These residuals and the logarithm of the number of observations were also used to test possible effects of feeding behaviour and life history by graph visualisation. Finally, we fit a generalised linear model using the *lm* function to test the effect of the logarithm of the distribution range of species interacting with IUCN threat status. This procedure also allowed us to understand the effect of the logarithm of the distribution range of species on the logarithm of the number of observations within each threat status group. For the R script with the codes for all statistical models described above see Supplementary Material 5.

**Open peer review.** To view the open peer review materials for this article, please visit http://doi.org/10.1017/ext.2024.22.

**Supplementary material.** The supplementary material for this article can be found at http://doi.org/10.1017/ext.2024.22.

**Data availability statement.** The raw data for this study are available at https://zenodo.org/record/7775610#.ZCHnGnbMLb0.

**Acknowledgements.** We thank all citizen scientists who contributed observations of birds in citizen science platforms that help biodiversity research of the Atlantic Forest and the observers who allowed us to use their photographs in the figures. We are grateful to two anonymous reviewers and the editor for their comments on earlier versions of this manuscript.

**Author contribution.** LRF conceived and designed the experiment, collected the data, performed data management, analysed the data, wrote the first draft of the manuscript, and approved the text; AMPRSP analysed the data, discussed the project, revised the manuscript and approved the text; TO, JL, AQ, MADFL collected the data, performed data management, discussed the project, revised the manuscript and approved the text; JKS discussed the project, collected the data, performed data management, wrote, edited and revised the manuscript and approved the text.

**Financial support.** This research received no specific grant from any funding agency, commercial or not-for-profit sectors.

**Competing interest.** None.

**Ethics statements (if appropriate).** This work did not require ethics approval.

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
