## [Editor Report]

Dear Dr. Szabo,

Thank you for submitting your manuscript for consideration by Cambridge Prisms - Extinction. Both reviewers were very positive about your study, but had a number of suggestions to improve the clarity of the manuscript. However, these suggestions constitute a minor revision in my opinion. I look forward to receiving a revised version that addresses their suggestions along with a detailed cover letter explaining how you addressed them. 

Best wishes,

Kate Lyons

---

## [Editor Report]

Dear Dr. Szabo,

Thank you for submitting your revised manuscript to Cambridge Prisms: Extinction. One of the previous reviewers has read the new version and has some concerns about the mixed effects models discussed in the text. There appear to be two separate issues. The first is that some analyses that are described as mixed effects models are actually linear models based on the submitted code. The second concerns the models fitted with an interaction. Some additional sensitivity analyses would be useful to evaluate the robustness of that model. 

I’d like to see a revised manuscript that clarifies the issues with the descriptions of the analyses in the methods and what is presented in the figures and also explores the robustness of the model with the interaction. 

Best wishes,

Kate Lyons

---

## [Editor Report]

Dear Dr. Szabo,

Thank you for submitting your revised manuscript for consideration by Cambridge Prisms - Extinction. I appreciate the efforts you have made to improve your manuscript in response to the reviewer comments. However, the reviewer still has concerns about the statistical methods and modeling that need to be addressed. I look forward to seeing a revised version along with a detailed cover letter that explains how you addressed the reviewer’s remaining concerns.

Best wishes,

Kate Lyons

---

## [Editor Report]

Dear Dr. Szabo,

Thank you for your efforts in responding to the reviewer comments. I am pleased to recommend your manuscript for publication in Cambridge Prisms-Extinction. 

Best wishes,

Kate Lyons